# Obesity and Overweight Are Associated with Minimal Extrathyroidal Extension, Multifocality and Bilaterality of Papillary Thyroid Cancer

**DOI:** 10.3390/jcm10050970

**Published:** 2021-03-02

**Authors:** Krzysztof Kaliszewski, Dorota Diakowska, Marta Rzeszutko, Jerzy Rudnicki

**Affiliations:** 1Department of General, Minimally Invasive and Endocrine Surgery, Wroclaw Medical University, Borowska 213, 50-556 Wroclaw, Poland; jerzy.rudnicki@umed.wroc.pl; 2Department of Nervous System Diseases, Faculty of Health Science, Wroclaw Medical University, Bartla 5, 51-618 Wroclaw, Poland; dorota.diakowska@umed.wroc.pl; 3Department of Pathomorphology, Wroclaw Medical University, Marcinkowski 1, 50-368 Wroclaw, Poland; rzemarta@wp.pl

**Keywords:** papillary thyroid cancer, obesity, overweight, extrathyroidal extension, multifocality, bilaterality

## Abstract

Epidemiological studies have shown a strong association between high body mass index (BMI) and papillary thyroid cancer (PTC). We assessed the clinical and histopathological features of PTC in patients with a higher BMI and compared them to analogous parameters in PTC patients with a normal BMI. We retrospectively analyzed 5021 medical records of patients admitted and surgically treated for thyroid tumors in one center between 2008 and 2018. Finally, we extracted data from 523 adult patients with PTC and stratified patients into two groups according to BMI: Group 1 with BMI < 25 kg/m^2^ and Group 2 with BMI ≥ 25 kg/m^2^. Data stratification was performed to estimate the association of overweight and obesity with clinical and histopathological features of PTC in both univariable and multivariable binary logistic regression analyses. Overall, compared to patients with a normal BMI, overweight and obese patients had a greater risk of minimal extrathyroidal extension (minimal ETE), multifocality and bilaterality of PTC (*p* < 0.001 for all). Analysis did not show that BMI was significantly associated with the more advanced tumor-node-metastasis (TNM) stage (*p* = 0.894). Obesity and overweight were significantly associated with higher aggressiveness of PTC. When considering various management options for PTC patients, these findings regarding overweight and obesity should be taken into consideration during the decision-making process.

## 1. Introduction

Obesity is defined as an excessive accumulation of body fat. The incidence of obesity has dramatically increased in recent decades and has become a major public health problem worldwide [1,2,3]. The prevalence of obesity tripled from 3.2% to 10.8% in men and doubled in women from 6.4% to 14.9% between 1975 and 2014 [4].

It was estimated that a high body mass index (BMI) is associated with a higher risk of some malignancies, such as breast, colon or pancreatic cancers [5,6,7]. Steele et al. [8] found that more than half of all cancers diagnosed in women and about a quarter of all cancers diagnosed in men have been correlated with being overweight and obese. Thus, in some countries, excess body weight is treated as the second most important and, fortunately, potentially modifiable risk factor for cancers [9]. Some authors additionally noticed that obesity has been associated with aggressive cancer histopathological features and unfavorable outcomes, especially in breast cancer [10]. Finally, studies have demonstrated a positive correlation between obesity and thyroid cancer (TC) [11,12,13]. In the opinion of many researchers, modifiable risk factors for TC have been documented; however, some of the accurate environmental factors that have contributed to the rising prevalence of TC remain controversial [14]. They suggest that the deleterious effect of obesity may differ according to cancer type. There are some meta-analyses that have reported a 25% higher risk of TC in an overweight population and a 55% higher TC risk in obese people than in individuals with normal BMI [1,15]. Schmid et al. [15] estimated that a 5 unit increase in BMI and a 0.1 unit increase in waist-to-hip ratio increased the risk of TC by 30% and 14%, respectively. They added that this increased risk is specific for papillary thyroid cancer (PTC), follicular thyroid cancer (FTC) and anaplastic TC but is not observed for medullary thyroid cancer [15].

After analyzing some studies, we noticed that the correlation between PTC and high BMI is not as obvious and is still deeply debated worldwide [16]. Due to biochemical and pathophysiological studies, some authors conclude that this association might be causal [17], but others say that it might be driven by the overdiagnosis of indolent TC entities [18]. These authors suggest that obese people use health services more frequently, so they have a higher opportunity to receive indolent PTC diagnoses [18]. Paradoxically, to make this issue more interesting, or even complex, some other authors recommend that people with excess body weight be encouraged to join cancer screening programs [9]. We presented our opinion concerning “cancer screening activity” in a previous study; however, we did not analyze PTC screening according to BMI [19].

Considering the facts that the prevalence of PTC is increasing in many parts of the world [17], that the same trend is observed for obesity [2,3,17] and that people with high BMI are more predisposed to PTC morbidity [12,20], we asked the following question: (A) could PTC in obese people present more aggressive features? Subsequently, we considered (B) whether a higher BMI may result in more frequent clinical manifestations and a higher prevalence of PTC.

We assessed the clinical and histopathological features of PTC in overweight and obese patients and compared them to analogous parameters in PTC patients with a normal BMI.

## 2. Materials and Methods

All procedures were conducted in accordance with the ethical standards of the institutional and/or national research committee and with the 1964 Declaration of Helsinki and its later amendments or comparable ethical standards. Our study protocol was approved by the Bioethics Committee of Wroclaw Medical University, Wroclaw, Poland (Signature number: KB-724/2020).

### 2.1. Study Design and Methods

We performed retrospective chart reviews of 5021 patients with thyroid tumors who were admitted to and surgically treated at the Department of General, Gastrointestinal and Endocrine Surgery of Wroclaw Medical University (Poland) between 2008 and 2018. Among the patients, 596 (11.87%) had thyroid malignancies, and of these patients, we selected 523 (10.41%) with PTC. Medical charts and pathology reports of each individual were reviewed. If the patient received 2 or more operations (incidental PTC was diagnosed and reoperation was needed due to not performing radical surgery), only the first one was included in the analysis. All of the patients were staged in accordance with the tumor-node-metastasis (TNM) staging criteria proposed by the American Joint Committee on Cancer (AJCC) (8th Edition) [21]. Preoperative thyroid ultrasonography, ultrasound guided-fine-needle aspiration biopsy (UG-FNAB) and cytological examinations were performed in all cases. All cytological specimens were evaluated and assessed according to the Bethesda System for Reporting Thyroid Cytopathology (TBSRTC) criteria [22]. The surgical tissue specimens were fixed in 10% buffered formalin and diagnosed histopathologically. Representative blocks were selected. A minimum of 5–8 blocks were taken from each lesion. Serial sectioning and careful cutting of the representative tissue sample was performed. A routine method of specimen processing was performed. The sections were cut into 4-µm-thick sections, from which conventional hematoxylin and eosin (H&E) staining sections were prepared. The H&E sections were evaluated by two experienced thyroid lesion pathologists to confirm the diagnosis, features of the tumor and extent of the malignant process. The variables included patient age at diagnosis, sex, BMI, tumor size, lymph node metastasis (LNM), distant metastasis at initial treatment, multifocality, bilaterality, number of foci, microcalcifications, tumor shape, margins, pathological vascularity, capsular and vascular invasion and minimal extrathyroidal extension (ETE). The exclusion criteria included secondary tumors, previous head and neck surgery, head and neck radiation exposure, and incomplete clinical or histopathological data. Next, we analyzed anthropometric features and compared them with histopathological characteristics. Some of them were considered pathognomonic for aggressive PTC entities [23].

### 2.2. Anthropometric Measurements

On the day of admission to the surgery department for thyroid operation, the height and weight of every patient were measured. The data on height and weight served to calculate BMI; thus, BMI was calculated using the following formula: the weight in kilograms divided by the square of the height in meters (kg/m^2^). Categorization of BMI followed the quartile definitions of the World Health Organization (WHO) BMI standard [24]. There were 4 patients (0.8%) who were underweight with a BMI < 18.5 kg/m^2^, 369 patients (70.5%) who had a normal weight with a BMI of 18.5–24.9 kg/m^2^, 51 patients (9.8%) who were overweight with a BMI 25–29.9 kg/m^2^, and 99 subjects (18.9%) who showed obesity with a BMI ≥ 30 kg/m^2^. Due to the significant differences in the number of patients in particular subgroups, we decided to create two study groups: Group 1 with BMI < 25 kg/m^2^ (*n* = 373) and Group 2 with BMI ≥ 25 kg/m^2^ (*n* = 150). Fasting blood glucose (FBG) was measured to explore the relationship between obesity and impaired fasting glucose. Additionally, serum thyroid-stimulating hormone (TSH) concentration was measured in all patients at the time of thyroidectomy, as subclinical hypothyroidism is reported frequently with obesity [25]. In our department, all patients that qualified for surgery were euthyroid; however, we decided to check TSH levels in the formed groups of patients.

### 2.3. Measured Outcomes

We examined the association between histopathological features of PTC in patients with overweight and obesity and compared them with analogous parameters in PTC patients with normal BMI.

### 2.4. Statistical Analysis

Statistica v.13.3 software (Tibco Software, Inc., Palo Alto, CA, USA) was used for statistical analysis of the data. Descriptive data are presented as the number of observations and percent (for qualitative variables) or as the mean, standard deviation (±SD) and 95% confidence interval (95% CI) for quantitative variables. Comparisons of categorical variables were performed by Pearson’s chi-square test or Fisher’s exact test, and analyses of continuous variables were performed by Student’s t test. Univariable and multivariable logistic regression analysis was used to estimate the impact of BMI ≥ 25 kg/m^2^ on the risk of the presence of disease severity indicators in PTC patients. A stepwise method of multivariable logistic regression analysis (with *p* < 0.05 as an entrance criterion and *p* > 0.1 as a removal criterion) was conducted. Age, serum TSH level and serum glucose concentration were used as confounding factors. The Kaplan–Meier method and log-rank test were performed to compare the distributions of disease-free survival and overall survival between patients with BMI < 25 kg/m^2^ and BMI ≥ 25 kg/m^2^. All tests were two-sided, and *p* < 0.05 was considered statistically significant.

## 3. Results

### 3.1. Baseline Characteristics of Patients with PTC

A total of 523 PTC patients were investigated. In Figure 1, we present the schematic flowchart of the study design (Figure 1).

### 3.2. Demographic, Clinical and Histopathological Characteristics of PTC Patients According to BMI

The comparison of demographic, clinical and pathological features in PTC patients with normal weight (BMI < 25 kg/m^2^) and with overweight + obese (BMI ≥ 25 kg/m^2^) is shown in Table 1. 

The mean age was significantly lower in the group with BMI < 25 kg/m^2^ than in the group with BMI ≥ 25 kg/m^2^ (*p* = 0.005). Serum TSH and FBG concentrations were significantly higher in patients with BMI ≥ 25 kg/m^2^ than in patients in the BMI < 25 kg/m^2^ group (*p* < 0.0001 for both). The frequency of females was similar in the two study groups (*p* = 0.724). We also constructed a multivariate logistic regression analysis model without sex as a confounding factor.

As shown in Table 1, the rate of diagnosis of thyroid malignancy before surgery and total thyroid resection was significantly higher in patients with BMI < 25 kg/m^2^ than in those with BMI ≥ 25 kg/m^2^ (*p* < 0.0001 for both), while the risk of reoperation significantly increased in patients who were overweight and obese (*p* = 0.0001). There was no significant association between pTNM stage and BMI groups (*p* = 0.894). Unexpectedly, the rate of tumor progression and distant metastasis presence was significantly higher in patients with BMI < 25 kg/m^2^ than in those with BMI ≥ 25 kg/m^2^ (*p* < 0.05 for both), although the rate of LNM was significantly higher in the group of patients with overweight and obesity (*p* = 0.008).

### 3.3. Selected Ultrasound and Histopathological Features with Respect to BMI. Association of BMI ≥ 25 kg/m^2^ with Disease Severity Indicator.

In Table 2, we show the results of comparative analyses of ultrasound and histopathological features with respect to BMI. Microcalcifications, hypoechogenicity, high vascularity, minimal ETE, capsular invasion, irregular tumor shape, multifocality, bilaterality and a high number of foci were significantly more common in patients with BMI ≥ 25 kg/m^2^ (*p* < 0.05 for all).

Odds ratios for disease severity indicators obtained in univariate logistic regression analyses confirmed these results (Table 3). However, the significant association of BMI ≥ 25 kg/m^2^ with PTC severity factors was corrected in the multivariate model of logistic regression analysis, where age, TSH and FBG concentrations were confounding factors. Finally, patients with overweight and obesity had a significantly higher risk of bilaterality, multifocality, high vascularity, extrathyroidal extension and irregular tumor shape (*p* > 0.05 for all) than patients with normal weight (Table 3). Analysis did not show that BMI ≥ 25 kg/m^2^ was significantly associated with LNM, microcalcifications, hypoechogenicity, or capsular or vascular invasion (*p* > 0.05 for all).

We created two BMI groups (group BMI < 25 kg/m^2^ vs. group BMI ≥ 25 kg/m^2^) as predictors of disease-free survival and overall survival during the observation period (2008–2018). However, there were no significant differences in disease-free survival (*p* = 0.561) or overall survival (*p* = 0.268) between these two study groups.

## 4. Discussion

Obesity and PTC incidence have become major public health problems worldwide [2,3,17]. Some authors have noticed that the detection rate of thyroid nodules is much higher in female individuals than in male individuals and that thyroid nodules are significantly associated with sex, age, BMI and metabolic-related diseases [26]. They also noticed that male sex and underweight might be protective factors for thyroid nodules, while obesity, hyperlipidemia, high blood glucose and metabolic syndrome were associated with an increased risk of thyroid nodules [26]. Considering the highlighted problem of overdiagnosis and overtreatment of TC, these authors recommend thyroid ultrasound examination in routine health check-ups for the early detection and treatment of thyroid nodules. However, despite the latest observations of overdiagnosis and overtreatment of PTC and some recommendations for screening examinations of overweight individuals, obesity very often has been treated as a factor associated with an increased risk of TC [11,12,17].

Currently, BMI is the most useful parameter describing overweight and obesity of patients. However, its clinical value is still debated. Firstly, it does not measure the body fat directly. Secondly, the composition of the body adipose tissue changes with age. Thirdly, the relationship between BMI and body fat is different for men and women. Some long-term observational studies assessed that patients assigned as overweight according to the BMI parameter had the same or even better health profile outcomes than those with the normal BMI [27]. All these discrepancies regarding BMI parameters should be taken under consideration in clinical analysis.

Some large, well-designed epidemiological and observational studies have shown new insights about exposures such as obesity, which may influence the development of PTC [28]. Although the underlying biochemical mechanisms are not clearly defined, some researchers suggest that obesity-related metabolic abnormalities, such as long-term chronic inflammation, insulin resistance or some adipokines, may contribute to the positive correlation between TC risk and obesity [12,14,29,30]. For instance, there are some studies that describe a positive correlation between a cysteine-rich secreted protein, known as progranulin, and obesity-associated cancers [31,32]. However, in addition to obesity being recognized as a risk factor for TC, the role of some proteins in this malignancy is not clearly established. Kwon et al. [20] noticed that serum adiponectin and progranulin levels did not present significant differences between patients with benign thyroid nodules and PTC; however, they estimated that there would be increased serum progranulin levels in patients with PTC, except microcarcinomas, and in patients with PTC with microscopic and gross ETE. Other authors have indicated specific obesity features that induce complex pathological changes, including hyperglycemia, hyperinsulinemia, hyperlipidemia, oxidative stress, inflammatory responses and the previously mentioned adipokines [14]. Therefore, after analysis of these studies, we see that modifiable risk factors for PTC are still deeply debated. Additionally, contrary to these findings, some authors did not confirm a causal role of obesity in the development of benign nodular thyroid disease or TC [18].

The association between obesity and the clinicopathological features of PTC is also debated. Thus, our hypothesis presented in the introduction section seems rational, especially since there is some evidence from retrospective analyses confirming that obesity is associated with more aggressive features of PTC, such as larger tumor size, ETE, more advanced tumor stage, and persistent disease following surgical treatment [33,34,35]. However, in contrast, there are some studies that have not confirmed these observations. Rahman et al. [17] noticed that the increased risk associated with overweight or obesity does not vary by histological subtype or the presence or absence of adverse cancer histopathological features [17]. They concluded that the greater risk of BRAF-mutated PTCs among those with high BMI suggests that the association may not merely reflect greater health care service use and indicates an independent relationship between obesity and clinically relevant TC [17].

Regardless of many views, we state that the detection of aggressive features of PTC is of high importance for making presurgical decisions, ranging from active surveillance, per hemithyroidectomy to total thyroidectomy followed by lymphadenectomy. In the present report, we apply another perspective: an anthropometric one. Multifocality, bilaterality and ETE were associated with high BMI and are among the most important PTC diagnostic markers that are also increasingly believed to increase the invasiveness of PTC. In this study, we evaluated the association between BMI and the invasiveness of PTC. Pathology with sectional histology was performed on all of the surgical materials. Currently, little is known about the differences in the histopathological features of PTC in patients with high and normal BMI. For this reason, we sought to systematically determine the association between BMI and some histopathological parameters. We paid particular attention to aggressive features of PTC, for which therapeutic dilemmas are the most emphasized. According to our analysis, PTC patients with a higher BMI should be qualified by surgeons for radical treatment due to a strong correlation with aggressive features of PTC. Furthermore, this is the first analysis in which these histopathological parameters were strongly associated with higher BMI. Thus, the findings may help clinicians make therapeutic decisions. Hence, we recommend that the presence of a high BMI should be reported in the specimen’s referral, which could result in greater meticulousness in the specimen and tumor evaluation by the pathologists. Furthermore, such individuals require watchful follow-up and refinement of therapeutic regimens during the postoperative period. Together, these data suggest that a high BMI further increases the risk of PTC invasiveness, including multifocality, bilaterality and ETE. We also noticed a significant correlation between some other invasive PTC features, such as LNM and vascular and capsular invasions. These observations raise the question of what biological mechanisms might be involved in the association between high BMI and PTC invasiveness. In concordance, microcalcifications are a common pathological feature of several cancers, such as breast cancer, prostate cancer and pancreatic cancer. [36,37,38,39]. We also noticed a significant correlation between this characteristic and higher BMI in patients with PTC. Li et al. [40] estimated that tumor tissue calcifications were positively correlated with the invasiveness of PTC. Moreover, they estimated that obesity promotes the risk of tumor invasiveness in PTC combined with tumor tissue calcifications. In our previous study, we estimated that microcalcifications were positively correlated with aggressive micro PTC (MPTC) entities [41]; however, in the current study, we additionally confirmed that this characteristic significantly correlated with higher BMI.

The results of our study are in agreement with others. Li et al. [42] confirmed that a higher BMI is significantly associated with aggressive clinicopathological features of PTC. They also noticed sex differences; the prevalence rates of obesity and aggressive clinicopathological features were significantly higher in men. Zhao et al. [43] reviewed medical charts of 10,844 consecutive patients with thyroid tumors undergoing partial or total thyroidectomy between 1993 and 2015. They estimated that for every five unit increase in BMI, the risk-adjusted odds of malignancy increased by 36.6%. Individuals who were obese or overweight were associated with a high risk of PTC compared to the risk among normal weight individuals. They did not notice any sex differences among the analyzed patients. They noticed that obesity was an independent predictor for multifocality, which we also confirmed in our study. They did not notice any correlations between LNM and BMI parameters. Subsequently, the authors showed that BMI was associated with high TNM stages; however, this association disappeared after adjusting for confounding factors. They concluded that obesity was significantly associated with the risk of PTC in a large operative population [43].

Our study has some limitations. It is a retrospective analysis, and access to some necessary specific details was limited, which caused the exclusion of some cases. This study was observational in nature, which makes it difficult to control for all potential confounding factors, including age, sex, smoking status or iodine intake. There was selection bias because patients included in this study underwent UG-FNAB and surgery, indicating that patients were not representative of the whole population. The analyzed data came from a single medical center, so the possibility of selection bias cannot be ruled out. We performed our study and made conclusions after an analysis of clinical and histopathological features, which only indirectly indicate PTC cases with potentially poor or excellent prognoses. We enrolled and analyzed a very large number of TC subjects in the study. Some aggressive histopathological features of TC did not show a significant correlation with obesity. This might be because our study included a relatively small number of individuals.

In summary, obesity and overweight are significantly associated with higher aggressiveness of PTC. The presence of a high BMI should be reported in specimen referral for more accurate analysis in terms of multifocality and bilaterality. Various management options exist for PTC patients, ranging from active surveillance to hemithyroidectomy to total thyroid resection followed by lymphadenectomy; the possibility of the higher aggressiveness of PTC in overweight and obese patients should be taken into consideration during the decision-making process.

## Figures and Tables

**Figure 1 jcm-10-00970-f001:**
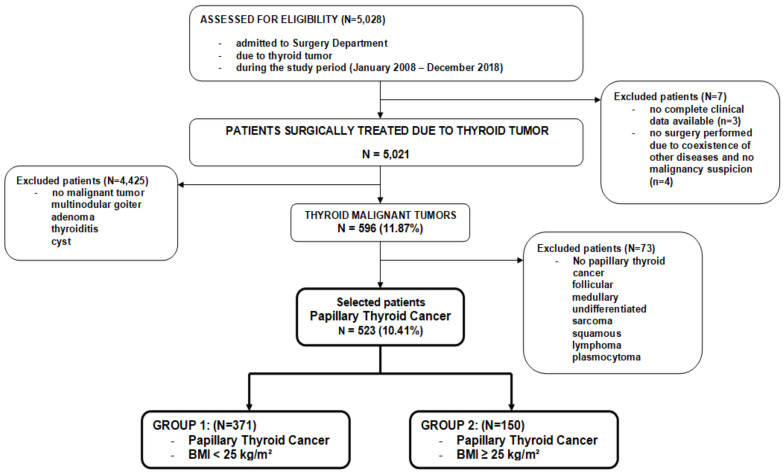
The schematic flow diagram of the study design; a total of 5021 patients diagnosed with thyroid tumors who underwent surgery were retrospectively reviewed in this study. Finally, 523 patients with papillary thyroid cancer (PTC) were enrolled and divided into two groups according to BMI.

**Table 1 jcm-10-00970-t001:** Demographic, clinical and pathological characteristics of the total group of PTC patients and two subgroups according to BMI. Data are presented as the number of observations (percent) or mean ± SD.

Parameters	Total	Group 1: BMI < 25 kg/m^2^ (*n* = 373)	Group 2: BMI ≥ 25 kg/m^2^ (*n* = 150)	*p*-Value
	(*n* = 523)			
Sex:				0.724
Female	456 (87.2)	324 (86.9)	132 (88.0)	
Male	67 (12.8)	49 (13.1)	18 (12.0)	
Age at diagnosis (years)	49.82 ± 15.67	48.59 ± 16.25	52.86 ± 13.73	0.005 *
Age:				0.127
<55 years	306 (58.5)	226 (60.6)	80 (53.3)	
≥55 years	217 (41.5)	147 (39.4)	70 (46.7)	
TSH (mU/L)	1.79 ± 1.14	1.45 ± 0.97	2.63 ± 1.08	<0.0001 *
Glucose (mmol/L)	4.86 ± 0.51	4.75 ± 0.49	5.16 ± 0.44	<0.0001 *
Diagnosis of thyroid malignancy:				<0.0001 *
Before surgery	317 (60.6)	247 (66.4)	70 (47.0)	
After surgery	206 (39.4)	125 (33.6)	79 (53.0)	
Type of surgery:				<0.0001 *
Radical	381 (72.8)	291 (78.0)	90 (60.0)	
Not radical	142 (27.2)	82 (22.0)	60 (40.0)	
Reoperation needed:				0.0001 *
No	388 (74.2)	294 (78.8)	94 (62.7)	
Yes	135 (25.8)	79 (21.2)	56 (37.3)	
pTNM stage:				0.894
I	433 (82.8)	307 (82.3)	126 (84.0)	
II	67 (12.8)	50 (13.4)	17 (11.3)	
III	15 (2.9)	10 (2.7)	5 (3.3)	
IV	8 (1.5)	6 (1.6)	2 (1.3)	
Tumor stage:				0.012 *
pT1a	209 (40.0)	136 (36.5)	73 (48.7)	
pT1b	238 (45.5)	181 (48.5)	57 (38.0)	
pT2	61 (11.7)	44 (11.8)	17 (11.3)	
pT3	7 (1.3)	7 (1.9)	0 (0.0)	
pT4a	3 (0.6)	3 (0.8)	0 (0.0)	
pT4b	5 (0.9)	2 (0.5)	3 (2.0)	
Tumor size:				<0.0001 *
<5 mm	89 (17.0)	66 (17.7)	24 (16.0)	
5–10 mm	120 (22.7)	70 (18.7)	49 (32.6)	
>10 mm	314 (60.3)	237 (63.6)	77 (51.4)	
Lymph node metastasis:				0.008 *
pN0	350 (66.9)	266 (71.3)	84 (56.0)	
pN1a	145 (27.7)	89 (23.9)	56 (37.3)	
pN1b	6 (1.1)	3 (0.8)	3 (2.0)	
pNx	22 (4.2)	15 (4.0)	7 (4.7)	
Distant metastasis:				<0.0001 *
pM0	450 (86.0)	337 (90.4)	113 (75.3)	
pM1	19 (3.6)	15 (4.0)	4 (2.7)	
pMx	54 (10.4)	21 (5.6)	33 (22.0)	

* statistically significant.

**Table 2 jcm-10-00970-t002:** Ultrasound and histopathological features of the total group of PTC patients and two BMI subgroups. Data are presented as the number of observations (percent).

Parameters	Total	Group1: BMI < 25 kg/m^2^	Group2: BMI ≥ 25 kg/m^2^	*p*-Value
	(*n* = 523)	(*n* = 373)	(*n* = 150)	
Microcalcification:				<0.0001 *
Yes	299 (57.2)	190 (50.9)	109 (72.7)	
No	224 (42.8)	183 (49.1)	41 (27.3)	
Echogenicity:				0.032 *
Hyperechoic	100 (19.1)	80 (21.4)	20 (13.3)	
Hypoechoic	423 (80.9)	293 (78.6)	130 (86.7)	
Vascularity:				<0.0001 *
High	269 (51.4)	166 (44.5)	103 (68.7)	
Low	254 (48.6)	207 (55.5)	47 (31.3)	
Extrathyroidal extension:				<0.0001 *
Yes	234 (44.7)	131 (35.1)	103 (68.7)	
No	289 (55.3)	242 (64.9)	47 (31.3)	
Capsular invasion:				<0.044 *
Yes	185 (35.4)	122 (32.7)	63 (42.0)	
No	338 (64.6)	251 (67.3)	87 (58.0)	
Vascular invasion				0.051
Yes	186 (35.6)	123 (32.9)	63 (42.0)	
No	337 (64.4)	250 (67.1)	87 (58.0)	
Tumor shape				<0.0001 *
Regular	255 (48.8)	207 (55.5)	48 (32.0)	
Irregular	268 (51.2)	166 (44.5)	102 (68.0)	
Sharp margins				<0.0001 *
Yes	250 (47.8)	206 (55.2)	44 (29.3)	
No	273 (52.2)	167 (44.8)	106 (70.7)	
Diagnosed as multifocal:				<0.0001 *
No	397 (75.9)	349 (93.6)	48 (32.0)	
yes	126 (24.1)	24 (6.4)	102 (68.0)	
Diagnosed as bilateral:				<0.0001 *
No	475 (90.8)	370 (99.2)	105 (70.0)	
Yes	48 (9.2)	3 (0.8)	45 (30.0)	
Number of foci:				<0.0001 *
1	397 (75.9)	349 (93.6)	48 (32.0)	
2	102 (19.5)	22 (5.9)	80 (53.3)	
3	24 (4.6)	2 (0.5)	22 (14.7)	

* statistically significant.

**Table 3 jcm-10-00970-t003:** Univariable and multivariable logistic regression analysis of BMI ≥ 25 as a risk factor for disease severity indicators in PTC patients. The multivariable logistic regression model was adjusted for age, serum TSH level and serum glucose concentration.

PTC Severity Indicators	Univariable Logistic Regression Model			Multivariable Logistic Regression Model		
	OR	±95% CI	*p*-Value	OR	±95% CI	*p*-Value
Advanced pTNM stage (III + IV)	1.09	0.43–2.72	0.849	-	-	-
pT progression (3 + 4)	0.61	0.17–2.21	0.452	-	-	-
Lymph node metastasis (pN1)	2.03	1.34–3.05	<0.001 *	1.41	0.86–2.31	0.168
Distant metastasis (pM1)	0.79	0.25–2.45	0.689	-	-	-
Bilaterality	52.85	16.05–173.90	<0.0001 *	22.76	6.39–81.08	<0.0001 *
Multifocality	30.90	18.0–52.9	<0.0001 *	20.07	9.54–42.24	<0.0001 *
Microcalcifications	2.56	1.69–3.87	<0.0001 *	1.65	0.99–2.75	0.051
Hypoechogenicity	1.77	1.04–3.02	0.034 *	1.46	0.79–2.71	0.224
High vascularity	2.73	1.82–4.08	<0.0001 *	1.9	1.20–3.03	0.006 *
Capsular invasion	1.49	1.00–2.20	0.045 *	1.07	0.67–1.72	0.768
Vascular invasion	1.47	0.99–2.17	0.052	1.06	0.66–1.71	0.795
Extrathyroidal extension	4.04	2.69–6.07	<0.0001 *	2.34	1.45–3.76	<0.001 *
Irregular tumor shape	2.64	1.77–3.95	<0.0001 *	1.87	1.18–2.97	0.007 *
Sharp margins	0.34	0.22–0.50	<0.0001 *	0.47	0.29–0.75	0.002 *

* statistically significant.

## Data Availability

The data are available from corresponding author and may be shared if necessary.

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
