# Peer review of "Obesity and Overweight Are Associated with Minimal Extrathyroidal Extension, Multifocality and Bilaterality of Papillary Thyroid Cancer"

_jcm, 2021, doi:10.3390/jcm10050970_

Round 1

Reviewer 1 Report

The authors have performed a nice retrospective study to analyze the association of BMI with the clinical and histopathological features of PTC. There are some limitations to the study, which have been discussed very well in the discussion section.

I have a few questions and suggestions:

  1. Do authors have data related to mutations (BRAF, RAS, TERT, etc.) in the analyzed cases? If yes, it will be interesting to include that information and perform correlation analysis to determine if the aggressive histopathological features observed in the cases with high BMI are associated with those mutations.
  2. Do authors have information on how many patients with high BMI had diabetes? Type 2 diabetes has been associated with increased thyroid cancer risk and may impact thyroid cancer aggressiveness.
  3. Did the authors analyze the lipid profile of the patients? If yes, that information should also be included.
  4. The discussion section seems to be quite elaborate and can be trimmed. 

Author Response

Journal of                                                                 February 3, 2021

Clinical Medicine

Dear Editor and Reviewers,

At the very beginning we would like to thank you very much for the possibility to re-submit our revised manuscript entitled Obesity and overweight are associated with minimal extrathyroidal extension, multifocality and bilaterality of papillary thyroid cancer. Thank you very much for considering it for potential publication in Journal of Clinical Medicine.

We would like to thank you for the very thorough reviews and for the advices and constructive criticism, which have been valuable for improving our paper. All of the suggestions for changes and improvements were very helpful to us, and we have revised the manuscript according to the recommendations made in the reviews. All of the changed and deleted portions of the manuscript are marked by using Track Changes. According to the reviewers’ instructions we corrected our manuscript point-by-point as follows.

We would like to thank the reviewer very much for the statements, that “we have performed a nice retrospective study” and that “some limitations (…) have been discussed very well in the discussion section.” Thank you very much.  However, regarding the concrete suggestions we answered them point-by-point as follows:     

Ad. 1 Dear reviewer, we performed retrospective analysis of the medical charts of the patients admitted and surgically treated at our surgical department. We analyzed all accessible data. Unfortunately, we did not find information about molecular tests of thyroid nodules such BRAF, RAS or TERT mutations. These tests are not routinely used in our country, so the patients did not undergo such analysis.

Ad. 2 Dear reviewer, we tried to analyze all of the clinical and pathological characteristics of the operated patients. However, the information about diabetes was not enough completed and sufficient for scientific analysis (type, level, duration, strong evidence of diagnosis), so we did not decide to include and analyze this parameter.

Ad. 3 Dear reviewer, the patients admitted to our department were prepared „strictly” and „only” for surgery, and we did not order any other additional biochemical parameters, which were not necessary to perform surgery.

Ad. 4 The discussion sections, which were quite elaborated were trimmed.

Dear reviewer,

thank you very much for your suggestions.

Thank you.

Reviewer 2 Report

This observational study of the association between papillary thyroid cancer (PTC) and overweight/obese patients is for the most part well written and the evidence, including case selection and statistical analyses, appears to be sound. I believe that the detailed information on tumor characteristics associated with aggressive malignancies associated with normal versus overweight/obese patients is a great contribution to the literature. As the authors note, the evidence of a causal link between obesity PTC epidemiology is equivocal. When random assignment is practically and ethically impossible, results from independent, observational studies help verify risk factors affecting disease epidemiology. This study contributes empirical evidence linking aggressive tumor characteristics to greater body mass. There are a few issues that need to be addressed, including a major ethical concern that must be considered. 

  1. A major concern is the reporting of cases (cell frequencies) in Tables 1 & 2. In the USA, data use agreements for HCUP, cancer registry population data sets, and the National Cancer Database specifically prohibit reporting cell frequencies when there are as few as 15 or fewer cases per cell. Even though these data may represent national samples and regarded as de-identified, the concern is that reporting too much detailed information among a small subset of cases places patient privacy at risk. There are a number of cells in Tables 1 & 2 that report fewer than 15 cases. There may be a different ethical standard where the data collection was performed for this study, or the journal may have different standards, but I would strongly urge the authors and journal editors to clarify if the tables conform to the journal's ethical standard regarding protected health information. The authors may want to consider reporting less detailed information for tumor characteristics, or collapsing categories to avoid risking patient privacy. If this recommendation is undertaken, the authors can easily add a note to the tables justifying this decision. Other published studies of relatively small samples of PTC or thyroid cancer may be referenced as examples of the use of "masking" data to protect patient privacy. 
  2. Related to #1, the authors would also need to delete information in the text of the manuscript that references very small numbers of cases, and replace frequencies in updated tables. 
  3. The authors refer to "reoperation" (e.g., line 82-83, 160) and "radical surgery" (line 245) several times in the manuscript  and also in Table 1 parameters (Type of surgery; Reoperation). It is unclear if they distinguish a "reoperation" from a "completion thyroidectomy." I'm uncertiain if thyroid surgery is practiced the same way where data collection was performed versus thyroid cancer surgery in the USA. I'll briefly note the practice of completion thyroidectomy as I know it. For some patients with thyroid neoplasms without indication of malignant tissue, a lobectomy or is planned prior to surgery based on ultrasound, needle biopsy, or other pre-op studies. After the procedure malignant tissue is identified, and a decision is made to remove the remaining thyroid lobe. This is described in records as either a "total thyroidectomy" or as a "completion thyroidectomy." They both represent total thyroidectomies, but the completion (2nd procedure) represents a 2-stage procedure. The second stage lobectomy typically occurs within a a short time or may be performed a few months after the first lobectomy. From my knowledge, endocrine surgeons label thyroid procedures performed 6 months or more after curative intent surgery as reoperations, and often in the context of persistent disease versus a recurrence.  Consequently, it is unclear if the authors regard a completion procedure as a reoperation, since they state that (line 82-83) only information from the first procedure was collected. How the authors define a "radical surgery" (line 83) is also unclear. Are procedures with lymph node resection, either completion or total thyroidectomy, categorized as "radical"? I would highly recommend that the authors I  review recent publications on thyroid cancer surgery to become more familiar with the definitions of total thyroidectomy, radical neck dissection, reoperation, and persistent versus recurrent PTC. The December issue of the journal "Surgery" includes manuscripts written by members of the American Association of Endocrine Surgeons and would help guide the authors toward use of these terms that conform to current international definitions. 
  4. In recent years, the use of BMI as a measure of overweight or obese health status is controversial. The authors may want to directly address this controversy in the Discussion section. 
  5. It is unclear to me if cases with micro-sized tumors were included or excluded from the data analyses? Related to this, I would find it helpful if authors added tumor size categories to either Table 1 or 2. Was it correlated with BMI? The authors refer to the association between tumor size and BMI on line 225, so it is unclear which variable in their study-analyses represents tumor size. 
  6. The remainder of my comments will relate to specific methodological or text-formatting issues. Line 127: I recommend adding "Pearson chi-square..."
  7. The authors use the terms "univariate" and "multivariate." Professional statisticians make a distinction between the definition of "multivariate" versus "multivariable." By these distinctions the authors analyses would more accurately be described as "univariable" and "multivariable." 
  8. Table 1, when printed out, some of the rows of numbers are out of alignment with text for "Parameters." My PDF printout for "Distant metastasis" the numbers of "pM0" appear between the line for "Distant mets..." and the line for "pM0", and the only numbers aligned with "pMx" are under the column for Group 2. This is just one example, so the table format should be reviewed and corrected. 
  9. Table 2, in the Parameters column "Diagnosed as multifocal" No/Yes, "yes" appears in lower case. 
  10. Table 2, in the Parameters column "Diagnosed as multifocal" and "Diagnosed as bilateral" the category line order is "No/Yes" but all prior variables appear in the order "Yes/No". Recommend consistent formatting. 
  11. Table 3, in the column for univariate 95% CI, numbers have 2 decimal places, except for "Multifocality". Recommend consistent formatting. 
  12. The last 2 lines of the Discussion (294-295) states that "More studies are needed...." This may be true, but it is a "boilerplate" or trite statement applicable to all observational studies. I would omit this line (the prior sentence is fine for ending the discussion), or write something more original. 
  13. References: it's something that may only annoy me, but for some references (#2, 7, 22, 24) all the words in title are capitalized, whereas all other references only the first word of the title is capitalized. I find it easier to read when the references use a consistent formatting. 
  14. References: it may be matter of journal formatting practices, but I believe any word appearing after a colon ":" should be capitalized. By this rule, the following references need correcting: 1, 4, 8, 10, 12, 13, 15, 16, 19, 21, 25, 26, 28, 36, 37, 38, 42. 
  15. References: It is unclear if #40 and #42 are chapters from a book? If so, I believe the title of the book and editor/authors should be included. 
  16. Reference #37: The abbreviation "MR" (as in MRI) appears in lower case. 
  17. References like #16, #17 appear to cite an electronic version of a Thyroid journal article. If the article has been published in print, I would include the issue and page numbers. 

Author Response

Journal of                                                                 February 3, 2021

Clinical Medicine

Dear Editor and Reviewers,

At the very beginning we would like to thank you very much for the possibility to re-submit our revised manuscript entitled Obesity and overweight are associated with minimal extrathyroidal extension, multifocality and bilaterality of papillary thyroid cancer. Thank you very much for considering it for potential publication in Journal of Clinical Medicine.

We would like to thank you for the very thorough reviews and for the advices and constructive criticism, which have been valuable for improving our paper. All of the suggestions for changes and improvements were very helpful to us, and we have revised the manuscript according to the recommendations made in the reviews. All of the changed and deleted portions of the manuscript are marked by using Track Changes. According to the reviewers’ instructions we corrected our manuscript point-by-point as follows.

We would like to thank the reviewer very much for the statements, that our “study (…) is for the most part well written and the evidence, including case selection and statistical analyses, appears to be sound.” We also thank for the opinion that “the detailed information on tumor characteristics associated with aggressive malignancies associated with normal versus overweight/obese patients is a great contribution to the literature.” This opinion is incredible important for us. Thank you very much. However, regarding the concrete suggestions we answered them point-by-point as follows:

Ad. 1 Dear reviewer, thank you very much for this remark. Indeed, because of the fact that some national cancer databases prohibit reporting cell frequencies when there are as few as 15 or fewer cases per cell, what we also noticed when we started to perform our analysis, and although, that we had one Bioethics Committee Approval for performing this analysis, we decided to apply for another one, in which we described this situation. We additionally attached our results and tables with our second application. After careful analysis, Bioethics Committee by our University Institution approved our study design with this situation as it stands and delivered new Bioethics Committee Approval (signature number KB-724/2020). We attached this new Bioethics Committee Approval. This approval is valid up to December 1, 2022. After this date, the rules probably will be changed in Poland. However now is valid and we got permission.   

Ad. 2 Dear reviewer, because we obtained new Bioethics Committee Approval with acceptation of this study design, with permission for publication of the clinical and pathological characteristics of our patients in the current numbers we would like to leave it as it stands.

Ad. 3 Dear reviewer, thank you very much for this remark. Although we thought, that our descriptions of the patients and procedures were clearly presented, we see that there might be some inaccuracies. We wrote, that (line: 80-83), „if the patient received 2 or more operations (….), only the first one was included in the analysis”. It is the only information that, even that some patient was admitted again to the Department for completion surgery (patient after lobectomy admitted for resection of the second lobe of the thyroid), he was not included to the study again as a new case of PTC. These patients (like this presented above) received „not radical” procedure. It means, that the patient was admitted to the clinic and underwent surgery due to thyroid nodule described before surgery as Bethesda I, II, III or IV, not malignant (V or IV). In such cases we were able to perform lobectomy with isthmectomy only. However, after histopathological analysis we received the histopathological report, that this was PTC with diameter above 1.0cm, so in such cases we qualified these patients to completion surgery. We named such situation in the table „not radical” We changed the term „partial” to „not radical”. During the second surgery, we performed total resection of the second, left lobe with central lymphadenectomy or lateral if needed. We also added in table 1 more clear term, instead of „Reoperation” we added „Reoperation needed: Yes No”.  

Ad. 4 Dear reviewer, indeed, in recent years, the use of BMI as a measure of overweight or obese health status is controversial, so we addressed this controversy in the Discussion section: „Currently, BMI is the most useful parameter describing overweight and obesity of patients. However, its clinical value is still debated. Firstly, it does not measure the body fat directly. Secondly, the composition of the body adipose tissue changes with age. Thirdly, the relationship between BMI and body fat is different for men and women. Some long-term observational studies assessed, that patients assigned as overweight according to BMI parameter, had the same or even better health profile outcomes than those with the normal BMI [27]. These all discrepancies regarding BMI parameter should be taken under consideration in clinical analysis.”

We added this reference and changed the numbers of the others.

  1. Mc Allister EJ, Dhurandhar NV, Keith SW. Ten putative contributors to the obesity epidemic. Crit Rev Food Sci Nutr. 2009; 49: 868–913.

Ad. 5 Dear reviewer, in our analysis we included all PTC, so also papillary thyroid microcarcinoma (PTMC). In table 1, in cell 9, we presented 209 cases of pT1a papillary thyroid microcarcinoma (PTMC), we compared the tumors’ stages (based on their size), and we obtained p value as statistically significant (p=0.012). When we formed our data base for this research, we obtained the information about accurate size of each microtumor. We divided them <5mm, 5-10mm. The PTCs with diameter above 10mm were 314 (not PTMC). We performed and included new statistical analysis according to tumor diameter (Table 1).

Ad. 6 According to reviewer’s recommendation, we changed “chi-square test” to  „Pearson’s chi-square test”. Thank you.

Ad. 7 We changed the terms „univariate” and „multivariate” for „univariable” and „multivariable.”

Ad. 8 Dear reviewer, thank you very much for this remark. We analyzed, reviewed and corrected all tables for their proper presentation. Thank you.

Ad. 9 We analyzed, reviewed and corrected table 2. Thank you.

Ad. 10 We formatted again all tables for their better presentation. Thank you.

Ad. 11 We reviewed, analyzed and formatted table 3, too. Thank you.

Ad. 12 Dear reviewer , we absolutely agree, that the last 2 lines of the Discussion (294-295) "More studies are needed...." is a "boilerplate" or trite statement applicable to all observational studies. We decided to remove it. Thank you for this remark.

Ad. 13 Dear reviewer, we changed all references to the „normal” standard and consistent formatting. However, as far as the reference number 22 Cibas et al., we would like to leave it as it stands, because the title of this article is de facto the name of the biopsy classification system known as TBSRTC: The Bethesda System for Reporting Thyroid Cythopathology used all over the world.

Ad. 14 Dear reviewer, we corrected all references according to journal formatting practice. Thank you.

Ad. 15 Dear reviewer, the references number 41 and 43 are articles published in journal „Medicine (Baltimore)” and we cited them according to PubMed base.

Ad. 16 We corrected this abbreviation form „mr” to „MRI”. Thank you.

Ad. 17 We added appropriate page numbers in these two articles:

  1. Park JH, Choi M, Kim JH, Kim J, Han K, Kim B et al. Metabolic syndrome and the risk of thyroid cancer: a nationwide population-based cohort study. Thyroid. 2020; 30: 1496-1504.
  2. Rahman ST, Pandeya N, Neale RE, McLeod DSA, Bain CJ, Baade PD et al. Obesity is associated with BRAF(V600E)-mutated thyroid cancer. Thyroid. 2020; 30: 1518-1527.

Dear reviewer,

thank you very much for this detailed, accurate and extremely helpful review.

Thank you.
